# The effect of cognitive behavioural therapy on pain and disability in chronic non-specific low back pain: An overview of systematic reviews

Tiffany Leung[1], Mei Ken Low[2], Pei Chi Yao[1], Ana-Catarina Pinho-Gomes[3]*

1 Institute of Epidemiology and Health Care, University College London, London, United Kingdom,
2 Institute of Ophthalmology, University College London, London, United Kingdom, 3 Institute for Global Health, University College London, London, United Kingdom

* a.pinho-gomes@ucl.ac.uk

## Abstract

### Background

Low back pain (LBP) is experienced by most people at least once in their lifetime and is the leading cause of disability worldwide. Cognitive behavioural therapy (CBT) is a non-invasive method used to manage chronic non-specific low back pain (CNSLBP) and contains less adverse effects than analgesic medications.

### Aims

To determine the efficacy of interventions incorporating CBT on reducing pain intensity and physical disability in CNSLBP.

### Methods

An overview of reviews without meta-analysis was conducted. Four electronic databases (MEDLINE, Embase, Cochrane Library and PsycINFO) were searched. Adults aged 18 or above with CNSLBP were included. AMSTAR 2 was used to assess risk of bias of included systematic reviews and data synthesis was conducted via vote counting methods.

### Results

Ten systematic reviews were included. Results showed that CBT is more effective in reducing pain and disability compared to waiting list/usual care (WL/UC). CBT in conjunction with other active treatments (AT) are also more effective than some stand-alone AT such as physiotherapy and exercise, but less effective than surgery.

### Conclusion

CBT has some effect in reducing pain and disability, however, it may not be more effective than other treatments. The quality of evidence was low for most outcomes

**Data availability statement:** All relevant data are within the paper and its Supporting information files.

**Funding:** The author(s) received no specific funding for this work.

**Competing interests:** The authors have declared that no competing interests exist.

as there was high risk of bias and heterogeneity between studies. Future research could evaluate specific therapies such as acceptance and commitment therapy (ACT) and mindfulness-based cognitive therapy (MBCT), as these treatments lacked primary studies. There is no recognised standard for conducting CBT, and further research could determine the most effective duration, mode of delivery (e.g., online, group-based) and intensity of CBT.

## Introduction

Low back pain (LBP) is a common medical condition that describes pain arising from the 12th ribs down to the hip and buttock area [1,2]. The predominant complaints about having LBP are pain and disability. LBP is the biggest cause of disability in the world, leading to a loss of workforce productivity [3]. People with LBP often have a lower quality of life and may suffer from depression due to inability to engage in regular work and social activities [4].

In 85–90% of people with LBP [5], this pain is not attributable to a specific patho-anatomical source – this is termed as non-specific [1,6]. It is speculated to be musculoskeletal in nature, arising from overuse or strain injury [5].

Most people experience LBP at least once in their lives [3]. It is estimated that in the UK, the 1-month period prevalence of LBP is 28.5%. It peaks between ages of 41–50 at 29.8%, though it affects people of all age groups. This falls to 25% at 80 years of age, however the severity of LBP increases with age [7]. According to the UK Office of National Statistics, there were more than 250 000 people of working age who were economically inactive due to back or neck problems in 2022 [8]. This results in both direct healthcare and indirect absent workforce costs to the economy, and is estimated to cost more than £10 billion [9].

As per National Institute for Health and Care Excellence (NICE) guidelines, current management methods for LBP consist of pharmacological, invasive and non-invasive treatments [10]. Pharmacological treatment focuses on reducing pain intensity, but have numerous adverse effects and reduced efficacy with repeated use [11]. Invasive treatments include back injections and surgery, but there is insufficient evidence supporting their utility [12]. These treatments are indicated for patients with sciatica who have failed to respond to non-surgical management. They are associated with procedural risks such as infection, bleeding and damage to local anatomical structures [13].

Non-pharmacological treatment methods include self-management, exercise, orthotics, manual therapies, acupuncture, electrotherapies, return to work programmes, psychological therapies and combined physical and psychological programmes [10]. The NICE guidelines offer limited guidance on the utilisation of psychological interventions, currently only recommending their use when combined with physical activity and underpinned by a cognitive behavioural approach.

Cognitive behavioural therapy (CBT) is the most well-known psychological therapy and is part of the "second wave" of behavioural therapy for managing psychiatric disorders. Whilst CBT does not remove the source of pain, it encourages a problem-solving

attitude by helping people develop strategies to alter their negative perception of pain and increase physical function [14,15]. By allowing people to adapt and cope better with discomfort, chronic pain patients develop increased pain tolerance and exhibit reduced limitation of physical activity. Many have reported feeling less pain and improved functionality [14,16].

More modern forms of CBT have been developed recently, bringing about the "third wave" of behavioural therapy. The "third wave" integrates other concepts such as metacognition, mindfulness, acceptance, personal values and goals into CBT [4,17]. Examples of the "third wave" include mindfulness-based cognitive therapy (MBCT) and acceptance and commitment therapy (ACT) [17]. With the advancement of psychological therapies, it is important to re-examine how treatments incorporating cognitive behavioural approaches affect LBP and how it compares to other management methods.

The existence of CBT spans a few decades, and a preliminary search revealed that there are numerous systematic reviews and literature reviews published on assessing the efficacy of CBT in managing LBP [18,19]. These studies conclude that CBT is effective in reducing pain and improving physical function in people with LBP. However, there is considerable heterogeneity in intervention duration, intensity and types of adjunct therapies used in combination with CBT across these studies.

To address this heterogeneity, an overview of reviews was conducted to synthesise evidence from multiple systematic reviews, generating a more comprehensive review of this topic. An overview of reviews is similar to a systematic review, with the difference in the unit of interest being systematic reviews instead of RCTs, non-randomised studies or observational studies [20]. This overview aims to determine the efficacy of interventions incorporating CBT, as well as identify areas where further research is needed, such as whether CBT can be standardised.

## Methods

This overview was written following the guidance of Cochrane Handbook of Systematic Reviews [21] and Synthesis without meta-analysis (SWiM) guidelines [22]. A protocol was submitted to PROSPERO (CRD42024527690) on 26th March 2024, and the PRISMA checklist is shown in S1 and S2 Checklist [23].

### Search strategy

A search of eligible systematic reviews was conducted in 4 electronic databases – MEDLINE, Embase, Cochrane Library and PsycINFO. Clinical trial registers were not applicable to this overview as the units of interest were systematic reviews instead of individual trials. The search was conducted on 20th April 2024 and included studies from the inception of each database through to that date. The search strategy was comprised of Medical Subject Headings (MeSH), keywords and phrases that were similar to "low back pain", "CBT", "systematic review". The full search strategy is highlighted in S3 Supporting Information.

### Eligibility criteria

**Population.** The study population was adults aged 18 years or older, with chronic non-specific low back pain (CNSLBP) lasting at least 3 months [24].

**Interventions.** Interventions were any therapies that incorporated a cognitive behavioural approach. This included standalone CBT and any treatment that contains a cognitive behavioural element. Examples of such treatments were multidisciplinary programmes, cognitive functional therapy (CFT), graded activity (GA), MBCT and ACT.

**Comparisons.** All comparison groups were eligible for inclusion and were divided into passive control or active treatment (AT). No treatment, sham treatment, waiting list (WL) or usual care (UC) were included in passive control. All other interventions were considered as AT.

**Outcomes.** The primary outcomes of interest were pain intensity and disability. At least one of the two must be a primary outcome in the systematic review to be eligible for inclusion. In some reviews, disability was referred to as physical function.

**Exclusion criteria.** Known inflammatory, infectious, neurological, or neoplastic causes of LBP and spinal injuries were excluded as it would not be classified as non-specific LBP. Pain duration of less than 3 months and population including individuals younger than 18 years old were not eligible. Systematic reviews that analysed mixed psychological interventions or mixed chronic pain together were not eligible.

## Study selection

The study screening was independently conducted by TL and ML. All disagreements were screened by a third reviewer, PY. Results of the search were deduplicated using EndNote 20 [25], following Jane Falconer's approach from London School of Hygiene and Tropical Medicine [26]. After deduplication, results were exported to Rayyan for title and abstract screening [27]. Texts eligible for full text screening were exported back to EndNote 20 [25].

The database search was not restricted by language; however, texts were excluded at full text screening if no English translation was found. For protocols that met the inclusion criteria, a search for a complete published review was conducted.

Multiple systematic reviews that used the same clinical trials would affect the results of an overview of reviews. A decision tool from Pollock et al. [28] was used to decide which systematic review to include. All primary studies were listed and entered into a citation matrix to identify any overlapping primary studies. The corrected covered area (CCA) was calculated, as per Pieper et al. [29]. CCA≤5% was considered a slight overlap and CCA≥15% was considered a high overlap. Systematic reviews that had 100% overlap of primary studies with other systematic reviews were excluded, following the guidance from Hennessy et al. [30]

## Data extraction

A custom Excel spreadsheet was created for data extraction, independently performed by TL and ML. In cases of discrepancies that could not be resolved by discussion, a third reviewer, PY, was consulted. All data for search strategy (search database, search date), publication details (number of studies included, publication year, funding and conflict of interest), participants (including age, gender), interventions and comparisons (duration, intensity, setting), outcomes, results (qualitative and quantitative), methodological quality assessment within each systematic review, and certainty of evidence were collected. Missing data was reported as such without further data synthesis.

## Risk of bias assessment of systematic reviews

The Risk of Bias assessment of included systematic reviews was independently conducted by TL and PY, using the AMSTAR 2 instrument from Shea et al 2017 [31]. Any discrepancies that were not resolved by discussion were settled by a third reviewer, ML. The rating of the overall confidence of results depended on the number of critical and non-critical domains satisfied.

## Data synthesis

Interventions were arranged in the following groups: standalone CBT, CBT in combination with AT, multidisciplinary programmes involving CBT and specific "third wave" therapies that incorporate cognitive behavioural elements such as CFT and MBCT.

A qualitative synthesis was conducted using a vote counting method based on direction of effect, following guidance from Cochrane Handbook [21] and Boon et al [32]. Vote counting was utilised to provide a transparent, reproducible and systematic method to compare the direction of results across multiple systematic reviews. This method aligns with the guidance from BMJ Synthesis Without Meta-analysis (SWiM) guidelines as a method to reduce subjectivity and possible biases that may be associated with a narrative synthesis [22]. Direction was reported if there were >70% of studies supporting treatment. If there were <70% of studies, this was considered indeterminate direction.

Other synthesis methods was not appropriate in this overview [21]. The heterogeneity in outcome measurements across different systematic reviews have made it difficult for quantitative synthesis. Moreover, not all systematic reviews provided an estimate of effect, precise p-value, or conducted a meta-analysis.

## Results

This overview of reviews included 10 systematic reviews, as seen in Fig 1 showing the PRISMA flow diagram [23]. The section "Methodological quality within individual systematic reviews" describes the quality of evidence from the RCTs assessed in those reviews. Moreover, the "Risk of bias assessment of systematic reviews" section presents the results of the AMSTAR2 assessment.

Data extracted from individual systematic reviews and data analysis via vote counting are classified into three categories: standalone CBT, CBT in combination with other treatments, and multi-disciplinary programmes.

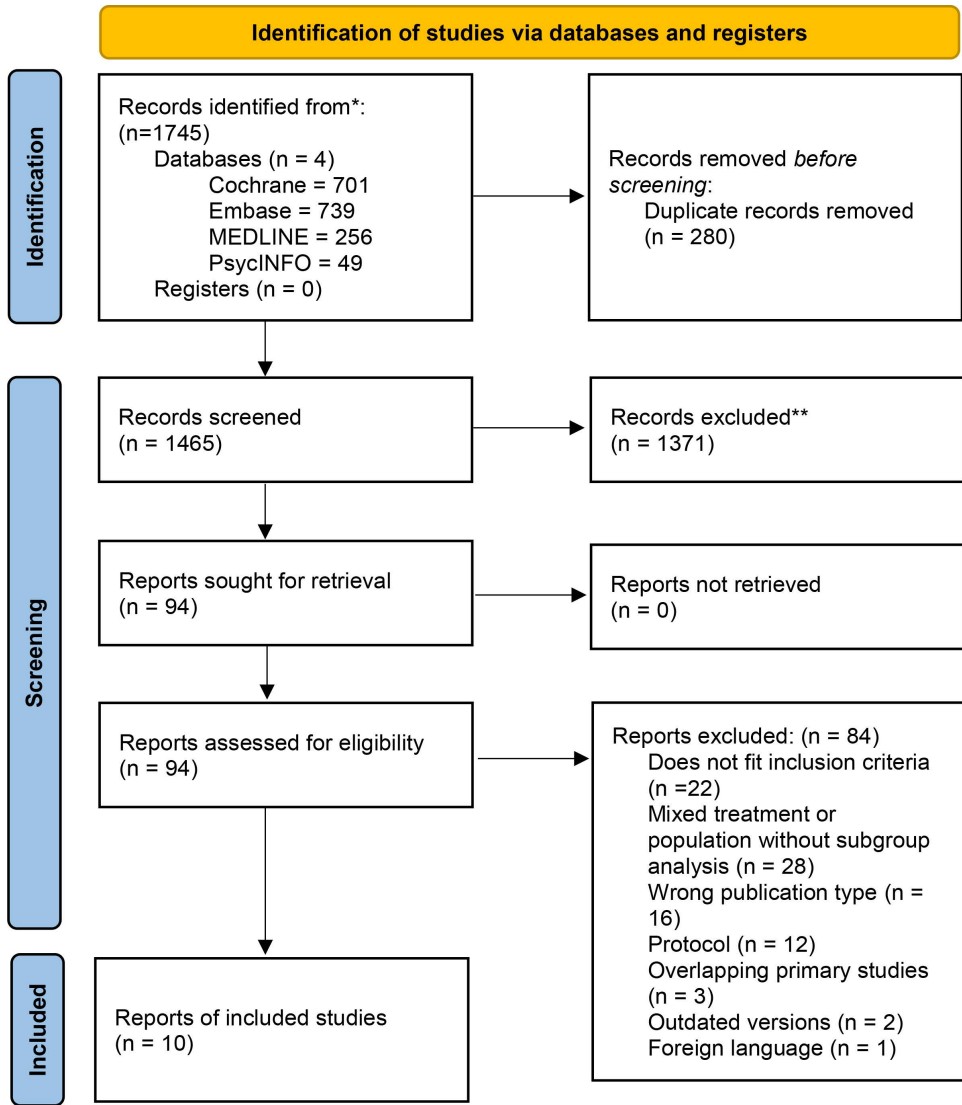

**Fig 1. PRISMA flow diagram.**

The search was conducted on 20<sup>th</sup> April 2024. Among the 94 full texts screened, 12 were protocols, of which 4 had full publications identified within the 94 full texts. Out of 82 full publications, 13 were eligible. The primary studies of the 13 systematic reviews were all RCTs. Non-Cochrane reviews were included in this overview as there was only one Cochrane review eligible for inclusion.

A citation matrix was done to identify overlapping primary studies, see S4 Table. CCA was 6.5%, indicating there is a low to moderate degree of overlap. Three of the eligible systematic reviews were found to have 100% overlap of primary studies with other systematic reviews and were subsequently excluded from analysis [33–35].

The full list of excluded studies is available in S5 Table.

Table 1 shows the characteristics of each individual systematic review.

## Methodological quality within individual systematic reviews

Table 2 summarises the methodological quality of individual studies within each systematic review. Seven systematic reviews used a risk of bias tool that was established by Cochrane [46,47]. More than half of the primary RCTs included are considered high risk or have some concern, where the most common domains of weakness were measurement of outcomes and lack of blinding in participants. Most RCTs had low risk in selection bias, randomisation process, allocation concealment and selective reporting.

Funding and competing interests are declared in all systematic reviews. Two systematic reviews had conducted their database search more than 2 years prior to publication.

## Results of individual systematic reviews

The results of the individual systematic reviews are summarised in table 3. The full results can be found in S6 Table.

## Standalone CBT

Most of the results showed that interventions with CBT were favoured over comparison groups, however they lack statistical significance. Few treatment methods such as respondent therapy, surgery and GEXP were more effective in reducing pain intensity and disability compared to interventions with CBT.

## CBT in combination with AT

Results showed that most of the comparisons favour CBT with AT, again lacking statistical significance similar to standalone CBT.

Two systematic reviews conducted multiple treatment comparisons. Ho et al. [40] conducted a network meta-analysis using the SUCRA mean rank to compare all psychological interventions in the management of CNSLBP. CBT with physiotherapy ranked first and is the most effective in reducing pain intensity in the long term, whereas standalone CBT ranked third in the long term. For reducing disability, CBT with physiotherapy ranked first immediately after treatment and long term, and CBT ranked sixth in long term.

Rihn et al. [43] conducted a likelihood of intervention superiority from multiple treatment analysis. Direct comparison between surgery and CBT with exercise showed both TDR and spinal fusion were more effective in reducing pain and disability in CNSLBP. Indirect comparison between physiotherapy and CBT with exercise demonstrated that physiotherapy was less effective than CBT with exercise in reducing pain and disability.

## Multidisciplinary programmes involving CBT

One systematic review examined multidisciplinary programmes. Jurak et al. [45] conducted a network meta-analysis using P-scores, where MBR-BE ranked first in reducing disability. MBR-BE ranked second in reducing pain.

**Table 1. Characteristics of included reviews.**

| Reviews | Participants | Intervention | Comparison | Primary Outcomes | Follow-up time points |
|---|---|---|---|---|---|
| Bahnamiri, 2020 [36] | 543 (5 studies)<br>Age: average 48.09 to 54.26<br>Gender: not available | Mindfulness-based interventions, such as MBSR, MBCT<br>90-120 min session for 8 weeks, with homework | UC;<br>CBT 90–120 min session for 8 weeks, with homework | Pain (BPI/NRS/MPQ) | Post intervention, 3, 6 and 12 months |
| Devonshire, 2023 [37] | 507 (5 studies)<br>Intervention: 262<br>Control: 245<br>Age: mean 44.6 (range 40.4–50.6)<br>Gender: 38% male | CFT – integrated behavioural approach led by physiotherapists to manage pain<br>30 min to 1 hour session, over 8–13.7 weeks<br>Delivered by physiotherapists | Physical therapy;<br>Exercise programmes | Pain (NRS/VAS)<br>Disability (ODI) | Post intervention, 6 and 12 months |
| Hajihasani, 2019 [38] | 965 (10 studies)<br>Intervention: 484<br>Control: 481<br>Age: range 18–65<br>Gender: not available | CBT + AT:<br>Exercise, physiotherapy or education<br>2-3 sessions per week over 3–12 weeks | Same AT as intervention group:<br>Exercise, physiotherapy or education | Pain (VAS/NRS/MPQ)<br>Disability (RMDQ/LBOS/LBP rating)<br>Quality of life, depression, functional capacity | Not available |
| Henschke, 2010 [39]<br>*Cochrane Review* | 2674 (21 studies)<br>Intervention: 1442<br>Control: 1035<br>Age: range 18–65<br>Gender: not available | CBT ± physiotherapy/rehabilitation programme over 3–12 weeks<br>Delivered at primary and secondary health centres | WL control; Exercise;<br>Other behavioural therapy;<br>Physiotherapy; Rehabilitation programme | Pain (VAS/NRS/MPQ)<br>Disability (RMDQ/ODI/QBPD/ADL/sickness impact profile) | Short (post intervention) Intermediate (6 months)<br>Long term (12 months) |
| Ho, 2022 [40] | 3851 (33 studies)<br>Intervention: 1792<br>Control: 2059<br>Age: mean 37.67 to 54<br>Gender: 8.33–65.68% male | CBT over 4–10 weeks;<br>CBT + physiotherapy over 3–52 weeks<br>Delivered in outpatient settings | Physiotherapy;<br>Other comparisons were not examined in network meta-analysis, including education, usual care, etc. | Pain (NRS/VAS/Box scale)<br>Disability (ODI/RMDQ/PDI) | Post intervention (0–2 months)<br>Short (2–6 months)<br>Mid (6–12 months),<br>Long term (≥12 months) |
| Lopez-de-Uralde-Villanueva, 2016 [41] | 1486 (12 studies)<br>Age: average 35–49<br>Gender: 25–58% male | GA – exercise using cognitive behavioural approach to improve activity tolerance by setting feasible goals at a rate manageable by participants | Exercise; Physiotherapy; WL;<br>crossover study with GEXP – similar to GA but exposes individuals to fearful situations instead of goal setting | Pain (VAS/NRS/MPQ)<br>Disability (RMDQ/QBPD/ PDI)<br>Pain catastrophising, quality of life | Short (<3 months)<br>Intermediate (3–12 months)<br>Long term (>12 months) |
| Petrucci, 2021 [42] | 1684 (13 studies)<br>Intervention: 883<br>Control: 801<br>Age: mean 39.61–64.3<br>Gender: 20–92% male | CBT ± exercise/medical treatment/education | UC; WL;<br>Same active treatment as intervention group | Pain (VAS/NRS/BPI)<br>Disability (RMDQ/ODI/ PROMIS)<br>Depression, anxiety | Mean 7.8 months, range from 3 weeks to 15 months |
| Rihn, 2017 [43] | 646 (4 studies)<br>Intervention: 328<br>Control: 318<br>In moderate to severe pain or disability (pain ≥ 5, ODI ≥ 30), degenerative changes in intervertebral discs at lumbar levels<br>Age: mean 39.4–44.1 Gender: 39–52% male | Spinal fusion;<br>Prodisc-L total disk replacement | Exercise + CBT for 60–110 hours over 3–5 weeks | Pain (VAS)<br>Disability (ODI), Complication risks, additional surgeries, cost-effectiveness | 12 and 24 months |

*(Continued)*

**Table 1.** (Continued)

| Reviews | Participants | Intervention | Comparison | Primary Outcomes | Follow-up time points |
|---|---|---|---|---|---|
| Yang, 2022 [44] | 3003 (22 studies) Age: not available Gender: 16.8–93.4% male | CBT; ACT; CBT + AT: UC, physiotherapy, exercise, rehabilitation programme Mean duration 10 weeks, 2–18 sessions lasting 30–120 mins each Majority are face to face, internet based, but also includes telephone, text and audiotape | UC; WL; Biofeedback; Same AT as intervention group: physiotherapy, exercise, rehabilitation programme, brief intervention | Pain (NRS/VAS) Disability (RMDQ/ODI/ADL) | Post intervention, 3, 6 and 12 months |
| Jurak, 2023 [45] | 1662 (18 studies) Age: average 44.61 (21.4 to 73.63) Gender: Male 31.18% | Multidisciplinary bio-psychosocial rehabilitation with behavioural, education or work-related components Mix of inpatient and outpatient | Minimal intervention; UC; exercise | Pain (VAS/MPQ/NRS) Disability (RMDQ/QBPD/ODI/PDI/Hanover functional ability) | Mostly 8 weeks, range from 2 to 12 weeks |

ACT, acceptance and commitment therapy; ADL, activity of daily living; AT, active treatment; BPI, brief pain inventory; CBT, cognitive behavioural therapy; CFT, cognitive functional therapy; GA, graded activity; GEXP, graded exposure therapy; LBOS, low back outcome scale; LBP, low back pain; MBCT, mindfulness-based cognitive therapy; MBSR, mindfulness-based stress reduction; MPQ, McGill pain questionnaire; NRS, numeric rating scale; ODI, Oswestry disability index; PDI, pain disability index; PROMIS, patient-reported outcomes measurement information system; QBPD, Quebec Back Pain Disability; RMDQ, Roland Morris Disability Questionnaire; UC, usual care; VAS, visual analogue scale; WL, waiting list.

## Certainty of evidence

GRADE was extracted from individual systematic reviews, seen in S6 Table. Not all systematic reviews assessed certainty of evidence. Six systematic reviews evaluated certainty of evidence, four of which used the GRADE tool [48]. Most reviews showed low to moderate certainty, where outcomes were ranked down in the inconsistency and imprecision domains. These were largely due to performance bias, large heterogeneity between RCTs and small sample sizes.

## Results of data synthesis

Data synthesis was conducted by vote counting of systematic reviews based on direction of effect. Table 4 shows the number of systematic reviews in each direction and overall direction for each comparison. The details of each comparison are shown in S7 Table.

## Standalone CBT

Table 4 shows that CBT, CFT and graded activity were more effective than WL/UC. The direction of effect for GA compared to AT was indeterminate. For comparisons involving CBT and AT, the direction of effect was more varied, and results were difficult to interpret as the specific treatments involved in each comparison were not clearly reported.

## CBT in co\mbination with AT

The addition of CBT to AT was found to be more effective than AT alone. However, 3 out of 4 systematic reviews in CBT with AT compared to same AT utilised physiotherapy as the active treatment, which is a potential source of bias.

**Table 2. Methodological quality within individual systematic reviews.**

| Reviews | Number of studies | Last searched | Funding and Competing Interests | Risk of Bias Tool | Risk of Bias Assessment |
|---|---|---|---|---|---|
| Bahnamiri, 2020 [36] | 5 | 2014 up to June 2020 | No funding, no conflicts of interest | Cochrane Risk of Bias tool (ROB 2) | 2 studies – low risk<br>3 studies – some concerns at randomization process, deviations from intended intervention and measurement of outcome |
| Devonshire, 2023 [37] | 5 | Up to March 2022 | No financial involvement with any organisations, no competing interests | Cochrane Risk of Bias 2 Tool | All studies – high risk<br>Measurement of outcome and selection of reported result |
| Hajihasani, 2019 [38] | 10 | Up to January 2018 | Support from Clinical Research Development Center of Rofeideh Rehabilitation Hospital and Neuromuscular Rehabilitation Research Center of Semnan University of Medical Sciences; authors disclosure: nothing to disclose | Hailey et al 2004 – Study assessment system | 7 studies – high quality<br>3 studies – good quality<br>Areas of bias include inadequate description of randomization methods and small sample size |
| Henschke, 2010 [39] *Cochrane Review* | 21 | Up to February 2009 | Multiple sources of support stated; 2 authors of this Cochrane review were co-authors of multiple included primary studies | Cochrane Back Review Group | 10 studies – low risk<br>11 studies – high risk<br>Majority of bias from blinding of participants and outcome assessors, inadequate randomization procedure and allocation concealment, inadequate compliance |
| Ho, 2022 [40] | 33 | Up to 31 January 2021 | No grant or financial relationship with any organisations, no competing interests | Cochrane Risk of Bias 2 tool | 3 studies – high risk<br>Majority of studies – some concerns in measurement of outcome and selection of reported results |
| Lopez-de-Uralde-Villanueva, 2016 [41] | 12 | Up to 10 December 2013 | No financial support, no conflicts of interest | PEDro scale | 8 studies – good methodology<br>2 studies – moderate methodology<br>2 studies – poor methodology<br>Areas of bias include lack of binding of participants, care provider and assessor |
| Petrucci, 2021 [42] | 13 | Last 25 years | Research grant from Italian Workers' Compensation Authority (INAIL), no conflicts of interest | Cochrane Risk of Bias tool | 9 studies – unclear<br>2 studies – high risk<br>2 studies – low risk<br>All had high risk for blinding of participants |
| Rihn, 2017 [43] | 4 | 1990 up to January 2014 | Grant from Association for Collaborative Spine Research, authors declared any financial payments or stock ownership interest | Journal of Bone & Joint Surgery criteria | All studies – moderately low risk<br>Areas of bias in blind assessment |
| Yang, 2022 [44] | 22 | 1980 up to 20 November 2021 | Research grant from National Natural Science Foundation of China, no conflicts of interest | Cochrane Back and Neck Review group | High risk of performance and detected bias |
| Jurak, 2023 [45] | 18 | Up to March 2022 | 1 author employed by "PhysioPlus" company, no financial support | Cochrane Risk of Bias 2 tool | 8 studies – high risk<br>5 studies – some concerns<br>5 studies – low risk<br>Areas of bias include deviation from intended intervention and measurement of outcome, few had bias in selection of reported results |

Mindfulness-based interventions (MBI) with CBT showed greater effectiveness than UC, but direction was indeterminate when compared to standalone CBT.

Surgery demonstrated superior outcomes in reducing pain intensity and disability compared to CBT with exercise. For the disability outcome between CBT with exercise and surgery, two systematic reviews relied on the same primary studies, introducing another potential source of bias.

**Table 3. Effect direction plot for results of individual reviews.**

| Reviews | Intervention/Comparison | Primary studies | Pain Intensity | Primary studies | Disability |
|---|---|---|---|---|---|
| Bahnamiri, 2020 [36] | CBT vs MBSR | 1 | ◄► | – | – |
| | MBCT vs CBT | 1 | ◄► | – | – |
| | Mindfulness + CBT vs Usual care | 3 | ▲ | – | – |
| Devonshire, 2023 [37] | CFT vs exercise/manual therapy | 5 | △ | 4 | △ |
| Hajihasani, 2019 [38] | CBT + physiotherapy vs physiotherapy | 5 | ▲ | 4 | ▲ |
| Henschke, 2010 [39] *Cochrane Review* | CBT vs Waiting list | 5 | ▲ | 4 | △ |
| | CBT vs cognitive therapy | 2 | △ | 2 | △ |
| | CBT vs operant therapy | 3 | △ | 2 | △ |
| | CBT vs respondent therapy | 3 | ▽ | 3 | ▽ |
| | CBT vs group exercise | 2 | ◄► | – | – |
| | CBT + exercise vs surgery | – | – | 2 | ▽ |
| | CBT + Physiotherapy vs Physiotherapy | 2 | △ | 2 | △ |
| | CBT + rehab vs rehab | 2 | △ | – | – |
| Ho, 2022 [40] | CBT vs Physiotherapy | 11 | △ | 8 | △ |
| | CBT + Physiotherapy vs Physiotherapy | 11 | ▲ | 11 | ▲ |
| Lopez-de-Uralde-Villanueva, 2016 [41] | GA vs other exercise | 4 | △ | 6 | △ |
| | GA vs Usual Care/Waiting List (control) | 4 | △ | 4 | ▲ |
| | GA vs GEXP | 2 | ▽ | 2 | ▼ |
| Petrucci, 2021 [42] | CBT vs control | 11 | ▲ | 9 | ▲ |
| | CBT vs MBSR | 2 | △ | – | – |
| Rihn, 2017 [43] | CBT + exercise vs TDR | 1 | ▽ | 1 | ▼ |
| | CBT + exercise vs fusion | 3 | ▽ | 3 | ▽ |
| Yang, 2022 [44] | CBT vs Usual Care/Waiting List | 5 | △ | 6 | △ |
| | CBT vs active treatment | 5 | △ | 4 | △ |
| | Concurrent CBT | 8 | ▲ | 9 | ▲ |
| Jurak, 2023 [45] | MBR-BE vs MI | 3 | ▲ | 3 | ▲ |
| | MBR-BE vs Usual care | 12 | ▲ | 12 | ▲ |
| | MBR-BE vs Exercise | 5 | △ | 5 | △ |

CBT, cognitive behavioural therapy; CFT, cognitive functional therapy; GA, graded activity; GEXP, graded exposure therapy; MBR-BE, multidisciplinary biopsychosocial rehabilitation – behavioural; MBCT, mindfulness-based cognitive therapy; MBSR, mindfulness-based stress reduction; MI, minimal intervention; TDR, total disc replacement; ▲, significant difference favouring intervention with CBT; △, non-significant difference favouring intervention with CBT; ▽, non-significant difference favouring comparison without CBT; ▼, significant difference favouring comparison without CBT; ◄►, no change/mixed findings.

## Multidisciplinary programmes involving CBT

Multidisciplinary programmes were more effective in reducing pain intensity when compared to AT and UC, however only two systematic reviews were involved in this comparison.

## Causes of heterogeneity

There was heterogeneity in population demographics, intensity and duration for each intervention and comparison. Furthermore, each primary study had different follow-up times and utilised different measurement scales for pain and disability. These were potential causes for heterogeneity in results.

**Table 4. Qualitative synthesis of results.**

| Interventions | Outcome | Favours intervention | Favours comparison | Indeterminate direction | Overall Direction |
|---|---|---|---|---|---|
| CBT vs WL/UC | Pain | 2 | 0 | 0 | ▲ |
|  | Disability | 2 | 0 | 0 | ▲ |
| CBT vs AT | Pain | 2 | 0 | 3 | ◄► |
|  | Disability | 3 | 0 | 1 | ▲ |
| CBT + exercise vs surgery | Pain | 0 | 1 | 0 | ▼ |
|  | Disability | 0 | 2 | 0 | ▼ |
| CBT + AT vs same AT | Pain | 4 | 0 | 0 | ▲ |
|  | Disability | 4 | 0 | 0 | ▲ |
| Multidisciplinary programme vs WL/UC | Pain | 1 | 0 | 0 | ▲ |
|  | Disability | – | – | – | – |
| Multidisciplinary programme vs AT | Pain | 2 | 0 | 0 | ▲ |
|  | Disability | – | – | – | – |
| GA vs WL/UC | Pain | 1 | 0 | 0 | ▲ |
|  | Disability | 1 | 0 | 0 | ▲ |
| GA vs AT | Pain | 0 | 0 | 1 | ◄► |
|  | Disability | 0 | 0 | 1 | ◄► |
| CFT vs exercise/manual therapy | Pain | 1 | 0 | 0 | ▲ |
|  | Disability | 1 | 0 | 0 | ▲ |
| Mindfulness + CBT vs UC | Pain | 1 | 0 | 0 | ▲ |
|  | Disability | – | – | – | – |
| Mindfulness + CBT vs CBT | Pain | 0 | 0 | 1 | ◄► |
|  | Disability | – | – | – | – |

AT, active treatment; CBT, cognitive behavioural therapy; CFT, cognitive functional therapy; GA, graded activity; WL/UC, waiting list/usual care; ▲, Studies overall favour intervention with CBT; ▼, Studies overall favour comparison without CBT; ◄►, no change/mixed findings.

### Risk of bias assessment of systematic reviews

The AMSTAR 2 tool [31] was used to assess the methodological quality of included systematic reviews. Table 5 shows the results of AMSTAR 2, where yellow boxes indicate a critical domain. Seven systematic reviews had low or critically low confidence. The most commonly missed critical domains were items 2 and 7, which would automatically result in critically low confidence. Multiple systematic reviews did not mention whether there were protocols registered or did not provide a list of excluded studies with justifications for exclusion.

Two systematic reviews did not carry out meta-analysis and could not be assessed on domains 11, 12 and 15. Regarding item 15, multiple studies evaluated publication bias as part of the GRADE assessment. As publication bias was assessed, they were counted as "yes", but their discussions contained little information about publication bias.

There are multiple reasons for the low quality of systematic reviews. There is less awareness and research on publishing high quality systematic reviews, as opposed to RCTs and observational studies. This may lead to many systematic reviews not addressing some of the critical domains mentioned. The first AMSTAR tool [49] was published in 2007, and the second updated version AMSTAR 2 was published in 2017. The changes between the two instruments may not be widely known, causing the systematic reviews to be assessed as low quality. Henschke et al [39] is a Cochrane review and scored moderate confidence, as it was first published in 2010, which was before AMSTAR 2 was published.

**Table 5. AMSTAR 2 quality assessment.**

| AMSTAR 2 Questions | Bahnamiri | Devonshire | Hajihasani | Henschke | Ho | Lopez-de-Uralde-Villanueva | Petrucci | Rihn | Yang | Jurak |
|---|---|---|---|---|---|---|---|---|---|---|
| 1. Did the research questions and inclusion criteria for the review include the components of PICO? | Yes | Yes | Yes | Yes | Yes | Yes | Yes | Yes | Yes | Yes |
| 2. Did the report of the review contain an explicit statement that the review methods were established prior to the conduct of the review and did the report justify any significant deviations from the protocol? | No | Yes | No | Yes | Yes | No | Yes | No | Yes | Yes |
| 3. Did the review authors explain their selection of the study designs for inclusion in the review? | No | No | No | No | No | No | No | Yes | No | No |
| 4. Did the review authors use a comprehensive literature search strategy? | Yes | Yes | Yes | Yes | Yes | Yes | Yes | Yes | Yes | Yes |
| 5. Did the review authors perform study selection in duplicate? | Yes | Yes | Yes | Yes | Yes | Yes | Yes | Yes | Yes | Yes |
| 6. Did the review authors perform data extraction in duplicate? | Yes | Yes | No | Yes | Yes | Yes | Yes | Yes | Yes | Yes |
| 7. Did the review authors provide a list of excluded studies and justify the exclusions? | No | Yes | No | Yes | Yes | No | No | No | No | No |
| 8. Did the review authors describe the included studies in adequate detail? | Yes | Yes | Yes | Yes | Yes | Yes | Yes | Yes | Yes | Yes |
| 9. Did the review authors use a satisfactory technique for assessing the risk of bias (RoB) in individual studies that were included in the review? | Yes | Yes | Yes | Yes | Yes | Yes | Yes | Yes | Yes | Yes |
| 10. Did the review authors report on the sources of funding for the studies included in the review? | No | Yes | No | No | Yes | No | No | Yes | No | No |
| 11. If meta-analysis was performed did the review authors use appropriate methods for statistical combination of results? | No meta-analysis | Yes | No meta-analysis | Yes | Yes | Yes | Yes | Yes | Yes | Yes |
| 12. If meta-analysis was performed, did the review authors assess the potential impact of RoB in individual studies on the results of the meta-analysis or other evidence synthesis? | No meta-analysis | No | No meta-analysis | No | Yes | Yes | No | Yes | Yes | Yes |
| 13. Did the review authors account for RoB in individual studies when interpreting/ discussing the results of the review? | Yes | Yes | Yes | Yes | Yes | Yes | Yes | Yes | Yes | Yes |
| 14. Did the review authors provide a satisfactory explanation for, and discussion of, any heterogeneity observed in the results of the review? | Yes | Yes | Yes | Yes | Yes | Yes | Yes | Yes | Yes | Yes |
| 15. If they performed quantitative synthesis did the review authors carry out an adequate investigation of publication bias (small study bias) and discuss its likely impact on the results of the review? | No meta-analysis | Yes | No meta-analysis | Yes | Yes | Yes | No | Yes | Yes | Yes |
| 16. Did the review authors report any potential sources of conflict of interest, including any funding they received for conducting the review? | Yes | Yes | Yes | Yes | Yes | Yes | Yes | Yes | Yes | Yes |
| Number of critical/non-critical weaknesses | 2/2 | 0/2 | 2/3 | 0/3 | 0/1 | 2/2 | 2/3 | 2/0 | 1/2 | 1/2 |
| Overall confidence in results rating | Critically low | Moderate | Critically low | Moderate | High | Critically low | Critically low | Critically low | Low | Low |

Yellow background, critical domain.

## Discussion

### Summary of results

Both standalone CBT and CBT with active treatment are more effective in reducing pain and improving disability compared to WL/UC. The evidence surrounding standalone CBT compared to AT is mixed, where some systematic reviews support CBT, however, the findings are not sufficient to establish a significant effect. CBT is more effective compared to physiotherapy, operant therapy and cognitive therapy, but less effective than surgery and respondent therapy.

CBT in combination with AT is more effective than AT itself. Specific treatments such as GA, CFT, MBI with CBT have limited evidence with only one systematic review each.

### Results in comparison to existing literature

As of the time of writing this overview, there is no published overview of reviews regarding CBT in CNSLBP. The results of systematic reviews that examined CBT as an adjunct therapy or comparing CBT to WL/UC demonstrated a significant effect in reducing pain and disability in CNSLBP, as per Mindouri et al [18].

The comparison between CBT and AT showed differing directions of effect, and this may be because the AT included were different treatments altogether.

Two systematic reviews identified that surgery was more effective than CBT and exercise together in reducing pain and improving physical function. This contrasts with current NICE guidelines, which recommend surgical interventions only for sciatica [10]. The discrepancy between our findings and current NICE recommendations may stem from the limited body of evidence available on the long-term effectiveness of surgical interventions for CNSLBP, as well as concerns regarding the potential risks and complications associated with surgical management. Ibrahim et al. reported that while surgical intervention led to only marginal improvements in disability outcomes, these effects were not statistically significant and were accompanied by an increased risk of surgical complications [50]. Although the study did not explicitly distinguish between non-specific and specific causes of low back pain, its findings align with the cautious approach of NICE. This overview's findings highlight important gaps in the current literature and suggests the need for further research evaluating the role and safety of surgical interventions in the management of CNSLBP.

### Overall completeness and applicability

Not all systematic reviews measured disability as an outcome. Specific forms of CBT such as CFT, GA and MBI with CBT only had one systematic review each. This highlights the need for further research in these areas. ACT and MBCT are some of the "third wave" behavioural therapy that could be included, however there is only 1 RCT by Godfrey et al. [51] involving ACT, and only 1 RCT related to MBCT by Day et al. [52]. All primary studies included were RCTs, which are considered as the gold standard for clinical trials. However, incorporating high-quality observational studies could offer complementary insights and help capture a more comprehensive understanding of the effects of the wide variety of interventions.

There is only 1 RCT conducted in 1997 by Rose et al. [53] that compared intensity, group and individual treatment. Further research is required in order to give recommendations for the most effective way of delivering CBT.

The systematic reviews include male and female genders, and age ranges between 18–65. However, the studies do not reflect how treatments affect people aged over 65 and populations from lower middle-income countries, as most of the studies are from higher income countries. This highlights possible areas of future research.

### Strengths

A thorough methodology was outlined and executed. The additional consideration of overlapping primary studies and qualitative data synthesis were performed via methods supported by literature on epidemiological research. There was

comprehensive data extraction of each individual systematic review. Results of both individual systematic reviews and data synthesis grouped by interventions were reported, adding to the current understanding of the effect of CBT on CNSLBP.

## Limitations

In this data synthesis, active treatment contains a mixture of treatments. Future research could explore the efficacy of individual interventions in isolation when compared to CBT. A meta-analysis would provide more precise information, including effect size, confidence intervals and statistics for heterogeneity.

There were multiple potential sources of bias in this overview of reviews. Grey literature was not searched, observational studies were not included, and only full texts of English papers were incorporated, leading to publication bias and language bias. A few systematic reviews that were included also reported having sources of funding or conflicts of interest, as mentioned in Table 2. Measurement bias is challenging to avoid due to the subjective nature of pain reporting and the lack of objective pain measurements.

75% of systematic reviews in CBT with AT compared to same AT utilised physiotherapy as AT, which could introduce bias as this would not be a good representation of all active treatments. When carrying out data synthesis, higher weighting could be applied for systematic reviews that have greater sample size and higher quality evidence. Ho et al. [40] had a large sample size of studies and low risk of bias, so should not be of equal weighting to other systematic reviews. Similarly, the overlap of primary studies in data synthesis should be considered. In the comparison for CBT with exercise against surgery, Henschke et al. [39] used the same RCTs as Rihn [43] and therefore the weighting of these studies should be lower.

There was high heterogeneity in study population. Some primary studies had a small sample size, leading to inadequate power of studies. Other systematic reviews have also commented on the lack of standardised outcome measures and different units of measurement that could alter the results [36,41,42,45].

Although the Cochrane Handbook states to use GRADE tool to assess certainty of evidence, there have been conflicts in literature for using GRADE in the context of Overview of reviews as it lacks formal guidance [54]. As such, the individual GRADE assessments from each systematic review were extracted as an alternative to conducting GRADE assessments.

## Conclusions

An overview of reviews was conducted to examine how CBT affects pain intensity and disability in CNSLBP. Systematic reviews suggest that CBT is most effective when integrated with other forms of active treatment, such as exercise or physiotherapy or as part of a multidisciplinary programme.

Implications for current clinical practice would be to continue providing treatment programmes that incorporate both physical and cognitive behavioural components. This would not only improve patient outcomes, but also restore a healthy workforce, and hence improve the economy. The availability of pain treatment programmes is often region specific, with variable length in waiting times [55]. The next step would be to evaluate the cost-effectiveness and make these programmes more widely available. Surgery was found to be more effective than CBT; however, this is not reflected in the current NICE guidelines [10], suggesting a potential area for further research.

There were also specific types of CBT that were not evaluated in this overview such as ACT and MBCT, which would again require further research in determining the efficacy of treatment. A method to standardise outcome measures would also facilitate comparisons across multiple studies.

Currently there is no agreed standard to the quality of care provided [55]. Future research may need to consider the optimal duration, intensity, and mode of delivery of CBT, including group, individual, telephone or internet-based treatment.

In conclusion, while this overview highlights the value of CBT particularly when combined with physical therapies in improving outcomes for CNSLBP, gaps in the current body of literature exist where more research is required.

## Supporting information

**S1 Checklist. PRISMA Checklist.**
(DOCX)

**S2 Checklist. PRISMA Abstract Checklist.**
(DOCX)

**S3 Supporting Information. Search Strategy.**
(DOCX)

**S4 Table. Citation Matrix of Primary Studies.**
(DOCX)

**S5 Table. Excluded Studies.**
(XLSX)

**S6 Table. Results of Included Systematic Reviews.**
(DOCX)

**S7 Table. Systematic reviews involved in each comparison.**
(DOCX)

## Author contributions

**Conceptualization:** Tiffany Leung, Ana-Catarina Pinho-Gomes.

**Formal analysis:** Tiffany Leung.

**Investigation:** Tiffany Leung, Mei Ken Low, Pei Chi Yao.

**Methodology:** Tiffany Leung.

**Supervision:** Ana-Catarina Pinho-Gomes.

**Writing – original draft:** Tiffany Leung.

**Writing – review & editing:** Mei Ken Low, Pei Chi Yao.

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
