## [Decision Letter · Decision Letter 0]

2 Jan 2025

Dear Dr. Pinho-Gomes,

We look forward to receiving your revised manuscript.

Kind regards,

Holakoo Mohsenifar

Academic Editor

PLOS ONE

**Journal Requirements:**

2. As required by our policy on Data Availability, please ensure your manuscript or supplementary information includes the following: 

Reviewers' comments:

Reviewer's Responses to Questions

**Comments to the Author**

1. Is the manuscript technically sound, and do the data support the conclusions?

Reviewer #1: Yes

Reviewer #2: Yes

2. Has the statistical analysis been performed appropriately and rigorously?

Reviewer #1: N/A

Reviewer #2: No

3. Have the authors made all data underlying the findings in their manuscript fully available?

Reviewer #1: Yes

Reviewer #2: Yes

4. Is the manuscript presented in an intelligible fashion and written in standard English?

Reviewer #1: Yes

Reviewer #2: Yes

**Reviewer #1: ** Thank you for the opportunity to review this study. I provide the following suggestions in hope to strengthen the work.

1. The introduction is clearly written and provides a detailed overview of low back pain and cognitive behavioural therapies. One important part of an introduction is to briefly describe the current literature on the topic (key findings, strengths, limitations) and then articulate how this study will contribute to the existing body of work.

2. Lines 67-69. I would reword your definition of non-specific low back pain to reflect it as a condition of exclusion of serious or specific pathologies. There may be pathology but it can not be attributed to a specific pathoanatomical source. I would also reword the second sentence to “…arising from an overuse or strain injury.” This will eliminate any controversy over the source of musculoskeletal pain, which could arise from spinal ligaments, discs, joints, etc - not just muscle. The reference by Bardin, King & Maher (2017) from Medical Journal of Australia may be helpful.

3. Lines 80-81. I would reframe this sentence as “… Invasive treatments include back injections and surgery, but there is insufficient evidence to support their utility”. You can reference the North American Spine Society (2020) guidelines for LBP. The NICE guidelines, which you reference, refer to spinal decompression only for sciatica which has not improved with conservative management. Spinal fusion and disc replacement are not recommended – this is important as it comes up in your results. Stating that invasive treatments is “…reserved for people with moderate to severe LBP” does not accurately reflect practice guidelines.

4. The study aim is not explicitly mentioned in the body of the text. It appears in the abstract only.

5. As you discuss the methodological quality/ risk of bias assessment, it is not always clear within the body of the text whether you are discussing the systematic reviews or the studies within the systematic reviews. For example, line 222 perhaps could read “… summaries the methodological quality of individual studies within each systematic review.” Perhaps subheadings could be better defined and consistent in the methods and results section so the reader clearly knows when you are discussion the risk of bias of systematic reviews vs studies within the systematic reviews. The tables are very clear.

6. Please provide further details about process of assessing methodological quality using the AMSTAR2 instrument. Did the reviewers work on this independently, and how were discrepancies managed?

7. Lines 205-206 “…12 protocols were included…” Perhaps use a different word to “included” as it confuses texts that met your inclusion criteria, when they were excluded.

8. Results are tabled really well. All abbreviations should appear in the legend, including CBT, CFT, GA, etc, even through these appear in the body of the text. There are many abbreviations throughout this paper. If the reader has difficulty recalling abbreviations, it will affect the flow of reading. Perhaps review which are the most important abbreviations. Some may be better written out if they appear infrequently or if they are not too long.

9. The results are presented under the headings: results of individual systematic reviews (line 233) and the results of data synthesis (line 271). The reporting appears a bit repetitive and jumps from interventions making it difficult to follow the results under these two headings. I wonder whether the results can be presented in a better order? Lines 185-187 present logical headings which you could report your findings under: standalone CBT, CBT in combinations, multi-d programs.

10. The discussion commences with a very clear summary of the study’s findings. This should be followed by discussing the key messages of the study and linking this to the literature. For example, one finding that warrants discussion is that spinal surgery is more effective than CBT. This is an interesting finding considering the NICE guidelines that you reference recommend against spinal surgery for chronic LBP. How your study’s findings compare to the known literature – whether in accordance or disagreement, needs discussion.

11. I am unclear why there are separate headings for potential bias (line 361) and limitations (392) in the discussion? Can these be reported under one heading?

12. Sentences that start with a number should be spelled out. Please consider this throughout the text.

**Reviewer #2:**  The whole paper requires some grammatical editing.

The data synthesis section needs revision: Why vote counting? In this context, in my opinion, narrative synthesis is more optimal than vote counting. Please provide more details

Some revisions in quality assessment section

**Do you want your identity to be public for this peer review?** For information about this choice, including consent withdrawal, please see our Privacy Policy

Reviewer #1: No

Reviewer #2: No

---

## [Author Response · Author response to Decision Letter 1]

25 Apr 2025

Dear Editors and Reviewers of PLOS One,

On behalf of all authors, I would like to thank you for reviewing our manuscript and the feedback provided. We have edited the manuscript accordingly. The amendments are listed below (the page and line numbers relate to the untracked copy):

Reviewer 1

1. The introduction is clearly written and provides a detailed overview of low back pain and cognitive behavioural therapies. One important part of an introduction is to briefly describe the current literature on the topic (key findings, strengths, limitations) and then articulate how this study will contribute to the existing body of work.

A paragraph has been added in the introduction to describe current literature and the limitations.

Page 5, lines 104-111

The existence of CBT spans a few decades, and a preliminary search revealed that there are numerous systematic reviews and literature reviews published on assessing the efficacy of CBT in managing LBP (18, 19). These studies conclude that CBT is effective in reducing pain and improving physical function in people with LBP. However, there is considerable heterogeneity in intervention duration, intensity and types of adjunct therapies used in combination with CBT across these studies.

To address this heterogeneity, an overview of reviews was conducted to synthesise evidence from multiple systematic reviews, generating a more comprehensive review of this topic.

2. Lines 67-69. I would reword your definition of non-specific low back pain to reflect it as a condition of exclusion of serious or specific pathologies. There may be pathology but it can not be attributed to a specific pathoanatomical source. I would also reword the second sentence to “…arising from an overuse or strain injury.” This will eliminate any controversy over the source of musculoskeletal pain, which could arise from spinal ligaments, discs, joints, etc - not just muscle. The reference by Bardin, King & Maher (2017) from Medical Journal of Australia may be helpful.

As per your suggestion, the definition of low back pain has been changed to ‘not attributable to a specific pathoanatomical source’ and ‘strain injury’. The reference from Bardin et al (6) has been used.

Page 3, lines 67-69

In 85-90% of people with LBP (5), this pain is not attributable to a specific pathoanatomical source – this is termed as non-specific (1, 6). It is speculated to be musculoskeletal in nature, arising from overuse or strain injury (5).

3. Lines 80-81. I would reframe this sentence as “… Invasive treatments include back injections and surgery, but there is insufficient evidence to support their utility”. You can reference the North American Spine Society (2020) guidelines for LBP. The NICE guidelines, which you reference, refer to spinal decompression only for sciatica which has not improved with conservative management. Spinal fusion and disc replacement are not recommended – this is important as it comes up in your results. Stating that invasive treatments is “…reserved for people with moderate to severe LBP” does not accurately reflect practice guidelines.

The sentence has been reframed as per your suggestions and the reference from NASS has been used (12). ‘…moderate to severe LBP’ has been changed to ‘sciatica which has not improved with non-surgical management’.

Page 3, lines 80-82

Invasive treatments include back injections and surgery, but there is insufficient evidence supporting their utility (12). These treatments are indicated for patients with sciatica who have failed to respond to non-surgical management.

4. The study aim is not explicitly mentioned in the body of the text. It appears in the abstract only.

The aim has been added to the end of the introduction, which is to determine efficacy of interventions incorporating CBT, and provide information for areas of potential future research.

Page 5, lines 114-115

This overview aims to determine the efficacy of interventions incorporating CBT, as well as identify areas where further research is needed, such as whether CBT can be standardised.

5. As you discuss the methodological quality/ risk of bias assessment, it is not always clear within the body of the text whether you are discussing the systematic reviews or the studies within the systematic reviews. For example, line 222 perhaps could read “… summaries the methodological quality of individual studies within each systematic review.” Perhaps subheadings could be better defined and consistent in the methods and results section so the reader clearly knows when you are discussion the risk of bias of systematic reviews vs studies within the systematic reviews. The tables are very clear.

The sentence describing table 2 has been changed as you suggested.

Page 15, line 250

Table 2 summarises the methodological quality of individual studies within each systematic review.

A paragraph has been added at the start of the results section to make the subheadings clear. Subheadings have now been changed and are consistent in methods and results section. Methodological quality within individual system systematic reviews refers to RCTs within the systematic reviews. Risk of Bias assessment of systematic reviews refers to the AMSTAR 2 tool.

Page 11, lines 223-226

The section “Methodological quality within individual systematic reviews” describes the quality of evidence from the RCTs assessed in those reviews. Moreover, the “Risk of bias assessment of systematic reviews” section presents the results of the AMSTAR2 assessment.

6. Please provide further details about process of assessing methodological quality using the AMSTAR2 instrument. Did the reviewers work on this independently, and how were discrepancies managed?

The methodology for using AMSTAR 2 has been described in detail. Two authors have conducted it independently. In the case of discrepancies that cannot be resolved with discussion between the first two, a third reviewer was consulted.

Page 9, lines 185-187

The Risk of Bias assessment of included systematic reviews was independently conducted by TL and PY, using the AMSTAR 2 instrument from Shea et al 2017 (31). Any discrepancies that were not resolved by discussion were settled by a third reviewer, ML.

7. Lines 205-206 “…12 protocols were included…” Perhaps use a different word to “included” as it confuses texts that met your inclusion criteria, when they were excluded.

The sentence has been rephrased and ‘included’ has been changed to ‘identified’.

Page 11, line 233

Among the 94 full texts screened, 12 were protocols, of which 4 had full publications identified within the 94 full texts.

8. Results are tabled really well. All abbreviations should appear in the legend, including CBT, CFT, GA, etc, even through these appear in the body of the text. There are many abbreviations throughout this paper. If the reader has difficulty recalling abbreviations, it will affect the flow of reading. Perhaps review which are the most important abbreviations. Some may be better written out if they appear infrequently or if they are not too long.

The abbreviations used in tables 1, 3 and 4 have all been listed out in the legend.

Page 14, Page 18 and Page 21

CBT, CFT, GA, LBP, MBCT have been written in full as cognitive behavioural therapy, cognitive functional therapy, graded activity, low back pain and mindfulness-based cognitive therapy respectively.

9. The results are presented under the headings: results of individual systematic reviews (line 233) and the results of data synthesis (line 271). The reporting appears a bit repetitive and jumps from interventions making it difficult to follow the results under these two headings. I wonder whether the results can be presented in a better order? Lines 185-187 present logical headings which you could report your findings under: standalone CBT, CBT in combinations, multi-d programs.

A paragraph has been added at the start of the results section explaining the changes. Data from individual systematic reviews and data synthesis are now grouped and discussed in 3 categories: ‘standalone CBT’, ‘CBT in combination with AT’ and ‘multidisciplinary programs involving CBT’. ‘Results of individual systematic reviews’ describes the results of each systematic review. ‘Results of data synthesis’ discuss results from vote counting.

Page 11, lines 227-229

Data extracted from individual systematic reviews and data analysis via vote counting are classified into three categories: standalone CBT, CBT in combination with other treatments, and multi-disciplinary programmes.

10. The discussion commences with a very clear summary of the study’s findings. This should be followed by discussing the key messages of the study and linking this to the literature. For example, one finding that warrants discussion is that spinal surgery is more effective than CBT. This is an interesting finding considering the NICE guidelines that you reference recommend against spinal surgery for chronic LBP. How your study’s findings compare to the known literature – whether in accordance or disagreement, needs discussion

The ‘Summary of results’ section is now immediately followed by ‘Results in comparison to existing literature’.

Page 27, line 387

A paragraph about surgery being more effective than CBT has been included, with some discussion about the discrepancies between our findings and NICE guidelines. These could be due to the risks of surgery outweighing the benefits. This has also been reflected in the conclusion.

Page 27, lines 394-406

Two systematic reviews identified that surgery was more effective than CBT and exercise together in reducing pain and improving physical function. This contrasts with current NICE guidelines, which recommend surgical interventions only for sciatica (10). The discrepancy between our findings and current NICE recommendations may stem from the limited body of evidence available on the long-term effectiveness of surgical interventions for CNSLBP, as well as concerns regarding the potential risks and complications associated with surgical management. Ibrahim et al. reported that while surgical intervention led to only marginal improvements in disability outcomes, these effects were not statistically significant and were accompanied by an increased risk of surgical complications (50). Although the study did not explicitly distinguish between non-specific and specific causes of low back pain, its findings align with the cautious approach of NICE. This overview’s findings highlight important gaps in the current literature and suggests the need for further research evaluating the role and safety of surgical interventions in the management of CNSLBP.

Page 31, lines 470-471

Surgery was found to be more effective than CBT; however, this is not reflected in the current NICE guidelines (10), suggesting a potential area for further research.

11. I am unclear why there are separate headings for potential bias (line 361) and limitations (392) in the discussion? Can these be reported under one heading?

The section for potential bias has been moved under limitations in the discussion, as you have suggested.

Page 29, lines 435-448

There were multiple potential sources of bias in this overview of reviews. Grey literature was not searched, observational studies were not included, and only full texts of English papers were incorporated, leading to publication bias and language bias. A few systematic reviews that were included also reported having sources of funding or conflicts of interest, as mentioned in table 2. Measurement bias is challenging to avoid due to the subjective nature of pain reporting and the lack of objective pain measurements.

75% of systematic reviews in CBT with AT compared to same AT utilised physiotherapy as AT, which could introduce bias as this would not be a good representation of all active treatments. When carrying out data synthesis, higher weighting could be applied for systematic reviews that have greater sample size and higher quality evidence. Ho et al. (40) had a large sample size of studies and low risk of bias, so should not be of equal weighting to other systematic reviews. Similarly, the overlap of primary studies in data synthesis should be considered. In the comparison for CBT with exercise against surgery, Henschke et al. (39) used the same RCTs as Rihn (43) and therefore the weighting of these studies should be lower.

12. Sentences that start with a number should be spelled out. Please consider this throughout the text.

The entire text has been checked for sentences beginning with a number and changes have been made accordingly.

Reviewer 2

1. The whole paper requires some grammatical editing.

The first, second and third author have proofread and made changes to the manuscript. The suggestions on the pdf (such as treat change to manage, and third “wave” change to “third wave”) have been followed.

2. Why vote counting? In this contex, in my opinion Narrative synthesis is more optimal than vote counting. Please provide more details

The paragraph describing vote counting has been expanded upon. It states that vote counting was used as a transparent, reproducible method to compare results across different systematic reviews.

Page 9, lines 193-198

A qualitative synthesis was conducted using a vote counting method based on direction of effect, following guidance from Cochrane Handbook (21) and Boon et al (32). Vote counting was utilised to provide a transparent, reproducible and systematic method to compare the direction of results across multiple systematic reviews. This method aligns with the guidance from BMJ Synthesis Without Meta-analysis (SWiM) guidelines as a method to reduce subjectivity and possible biases that may be associated with a narrative synthesis (22).

3. In overview of reviews we need quality assessment of systematic reviews. Please provide quality assessment section and write about methodological quality and risk of bias of SRs

The AMSTAR 2 tool has been used to assess risk of bias in each of the systematic reviews, outlined in table 5. The subheadings have been revised to provide more clarity that the risk of bias assessment has been conducted for each of the systematic reviews.

Page 22, line 342-344 and Page 24-25

The AMSTAR 2 tool (31) was used to assess the methodological quality of included systematic reviews. Table 5 shows the results of AMSTAR 2, where yellow boxes indicate a critical domain. Seven systematic reviews had low or critically low confidence.

4. I prefer to read about systematic reviews not the original articles. Please provide more detail about each systematic reviews.

The results of the individual systematic reviews have been outlined in table 3, and also described in the text by after classifying into “standalone CBT”, “CBT in combination with AT” and “Multidisciplinary programmes involving CBT”. The paragraph about the original RCT by Rose et al has been removed.

Page 18

The results of data synthesis via vote counting has been described in the following section.

Page 20, line 303

Data synthesis was conducted by vote counting of systematic reviews based on direction of effect. Table 4 shows the number of systematic reviews in each direction and overall direction for each comparison. The details of each comparison are shown in S7 Table.

Thank you once again for all your suggestions. We believe that the amendments made are satisfactory.

Yours sincerely,

Tiffany Leung

---

## [Decision Letter · Decision Letter 1]

7 May 2025

The effect of cognitive behavioural therapy on pain and disability in chronic non-specific low back pain: an overview of systematic reviews

PONE-D-24-51702R1

Dear Dr. Ana-Catarina Pinho-Gomes,

We’re pleased to inform you that your manuscript has been judged scientifically suitable for publication and will be formally accepted for publication once it meets all outstanding technical requirements.

Kind regards,

Holakoo Mohsenifar

Academic Editor

PLOS ONE

Additional Editor Comments (optional):

Reviewers' comments:

Reviewer's Responses to Questions

**Comments to the Author**

Reviewer #1: All comments have been addressed

Reviewer #2: All comments have been addressed

2. Is the manuscript technically sound, and do the data support the conclusions?

Reviewer #1: Yes

Reviewer #2: Yes

3. Has the statistical analysis been performed appropriately and rigorously?

Reviewer #1: N/A

Reviewer #2: Yes

4. Have the authors made all data underlying the findings in their manuscript fully available?

Reviewer #1: Yes

Reviewer #2: Yes

5. Is the manuscript presented in an intelligible fashion and written in standard English?

Reviewer #1: Yes

Reviewer #2: Yes

Reviewer #1: Well done on producing this manuscript. I am satisfied that the revision has adequately addressed all my feedback.

Reviewer #2: (No Response)

**Do you want your identity to be public for this peer review?** For information about this choice, including consent withdrawal, please see our Privacy Policy

Reviewer #1: No

Reviewer #2: **Yes: ** Sanaz Bemani

---

## [Editor Report · Acceptance letter]

PONE-D-24-51702R1

PLOS ONE

Dear Dr. Pinho-Gomes,

I'm pleased to inform you that your manuscript has been deemed suitable for publication in PLOS ONE. Congratulations! Your manuscript is now being handed over to our production team.

Kind regards,

on behalf of

Dr. Holakoo Mohsenifar

Academic Editor

PLOS ONE